# Synthesis and the In Vitro Evaluation of Antitumor Activity of Novel Thiobenzanilides

**DOI:** 10.3390/molecules28041877

**Published:** 2023-02-16

**Authors:** Maria João Álvaro-Martins, Violeta Railean, Filomena Martins, Miguel Machuqueiro, Rita Pacheco, Susana Santos

**Affiliations:** 1Centro de Química Estrutural, Institute of Molecular Sciences, Faculdade de Ciências, Universidade de Lisboa, Campo Grande, 1749-016 Lisboa, Portugal; 2Departamento de Química e Bioquímica, Faculdade de Ciências, Universidade de Lisboa, Campo Grande, 1749-016 Lisboa, Portugal; 3BioISI-Biosystems & Integrative Sciences Institute, Faculdade de Ciências, Universidade de Lisboa, Campo Grande, 1749-016 Lisboa, Portugal; 4Departamento de Engenharia Química, Instituto Superior de Engenharia de Lisboa, 1959-007 Lisboa, Portugal

**Keywords:** cancer, thiobenzanilide derivatives, melanoma, breast cancer

## Abstract

Cancer is a generic term for a large group of diseases that are the second-leading cause of death worldwide, accounting for nearly 10 million deaths in 2020. Melanoma is a highly aggressive skin tumor with an increasing incidence and poor prognosis in the metastatic stage. Breast cancer still stands as one of the major cancer-associated deaths among women, and diagnosed cases are increasing year after year worldwide. Despite the recent therapeutic advances for this type of cancer, novel drugs and treatment strategies are still urgently needed. In this paper, the synthesis of 18 thiobenzanilide derivatives (17 of them new) is described, and their cytotoxic potential against melanoma cells (A375) and hormone-dependent breast cancer (MCF-7) cells is evaluated using the MTT assay. In the A375 cell line, most of the tested thiobenzanilides derivatives showed EC_50_ values in the order of μM. Compound **17** was the most promising, with an EC_50_ (24 h) of 11.8 μM. Compounds **8** and **9** are also interesting compounds that deserve to be further improved. The MCF-7 cell line, on the other hand, was seen to be less susceptible to these thiobenzanilides indicating that these compounds show different selectivity towards skin and breast cancer cells. Compound **15** showed the highest cytotoxic potential for MCF-7 cells, with an EC_50_ (24 h) of 43 μM, a value within the range of the EC_50_ value determined for tamoxifen (30.0 μM). ADME predictions confirm the potential of the best compounds. Overall, this work discloses a new set of thiobenzanilides that are worth being considered as new scaffolds for the further development of anticancer agents.

## 1. Introduction

According to the World Health Organization, cancer is the second leading cause of death globally, being estimated in 2018 as the cause of death of 9.6 million people (one in every six deceases). Lung and breast cancers are the most common cancers worldwide, with an incidence of 2.09 million cases each [1].

Breast cancer is the most prevalent cancer in women and its incidence has been rising in the last decade [2]. Skin cancer is the 5th most incidence cancer in men and women, with over 1 million diagnoses worldwide in 2018, although this is likely an underestimated number. Melanoma is the least common but the deadliest skin cancer, accounting for 1% of all cases. While it is less frequent than basal cell carcinoma and squamous cell carcinoma, melanoma, which begins in melanocytes, is more dangerous because of its ability to rapidly spread to other organs if not treated at an early stage [3]. Whereas, treating early-stage melanomas only requires, in most cases, surgery; for metastatic melanomas that have spread beyond the skin, more aggressive approaches are involved. Surgery, chemotherapy and radiotherapy, the most common therapies available in the past, showed minor success in widespread melanomas, but recent advances in the so-called new therapies, including immunotherapy and targeted drugs, increased the five years’ survival rate of these patients by 52% [4], which is, however, still a very disappointing number. Targeted drugs act on specific genes or proteins involved in the growth and survival of cancer cells and are very helpful in treating melanomas that have certain gene changes, without as much damage to healthy cells as other treatment options such as chemotherapy [5]. There are a variety of gene mutations that can lead to metastatic melanomas, the most common one being the BRAF mutation (40%). This mutation affects the BRAF gene which is, in fact, part of the MAPK (Mitogen-Activated Protein Kinase) pathway, the signaling pathway that regulates intracellular processes, namely acute hormone responses, embryogenesis, cellular differentiation, proliferation and apoptosis [6]. This cascade pathway includes signaling proteins called Ras, Raf, MEK and ERK. Potent and specific inhibitors of MAPK pathway mediators have been discovered and include BRAF inhibitors, e.g., vemurafenib or encorafenib, and MEK inhibitors, e.g., trametinib (Figure 1), which are FDA- and EMA-approved drugs that have been used alone or in combination regimes against metastatic melanoma [7,8].

All these drugs have structural motives like carbo- or heteroaromatic moieties, especially aniline groups substituted with halogens (fluorine, iodine or chorine), and carbonyl groups that are thought to ensure the binding to the target protein [9]. Although efficient in the treatment of some patients, a significant resistance problem has been reported that can occur by several distinct and unpredictable mechanisms, even when these drugs are used in combination therapy, mitochondrial oxidative stress being one of them [10,11]. The advances in immunotherapy, alone or in combination with other therapies, have significantly changed the approach to treating melanoma [12]. Indeed, there are great advantages to using immunotherapy to treat cancer, since it uses the person’s own immune system to help in killing cancer cells. However, serious disadvantages can be associated with this type of treatment, mainly negative regulations leading to autoimmune diseases (sometimes chronic) and even death, several nonspecific toxic side effects and hyper-progressive disease occurrence, accelerating patients’ death [13,14].

These problems steer new metastatic melanoma treatment strategies, either through the development of other inhibitors for the already known targets, that may offer improved efficacy and tolerability, by searching for new targets [15] or even by seeking completely new approaches [16].

Thiobenzanilides have received considerable attention from the scientific community due to their wide range of pharmacological effects such as spasmolytic [17], antifungal [18], antibacterial [19,20] and anticancer [21,22] actions. Hu et al. [21] have studied for the first time the anticancer properties of thiobenzanilide derivatives on human melanoma A375 cells (Figure 2), finding that they all induced antiproliferation, leading to cell growth cycle perturbations, aberrant mitochondrial function and subsequent apoptotic cell death.

More recently, Kucinska et al. [22] have also studied a series of thiobenzanilide derivatives having found that three of them had antiproliferative activity against estrogen-dependent MCF-7 breast cancer cells and that, in addition, one of them was selective for those cells, being almost innocuous towards normal breast cells (Figure 3). Despite the fact that drugs approved for breast cancer treatment are efficient to inhibit cancer progression by a variety of different mechanisms, chemoresistance is still an important issue and one of the major challenges that patients have to face [23], justifying *per se* the search for new molecules with new targets or new mechanisms of action.

With this background in mind, we herein report the synthesis of a library of novel thiobenzanilide derivatives with structural features in both the acyl and aniline moieties, modified using several substitution patterns and substituents. In this work, the activity of all compounds was evaluated using in vitro models of human melanoma, namely cell line A375, and of MCF-7 human breast cancer cells.

## 2. Results and Discussion

### 2.1. Synthesis and Characterization of the Thiobenzanilide Derivatives

The new thiobenzanilides to be synthesized were chosen in such a way as to systematically vary the substitution pattern of the acyl and aniline moieties. Thus, for the same set of substituents in position 4 of the aniline moiety (CH_2_Ph, OPh, (CH_2_)_7_CH_3_, O(CH_2_)_7_CH_3_ and H) three substitution patterns in the acyl group were chosen: no-substitution, substitution at position 2 (with either a CF_3_ or a NO_2_ group) and di-substitution in positions 3 and 5 (with NO_2_ groups). Substituents CF_3_ and NO_2_ were selected based on results previously obtained by Hu et al. [21]. Both substituents have an electron-withdrawing nature, in the case of the nitro group through a mesomeric effect, while in CF_3_ through an inductive effect which is predominant.

Compounds **1**–**18** (Table 1) were prepared by a well-known synthetic route. In the first step, the Schotten–Baumann reaction between the appropriate acid chlorides and amines in the presence of pyridine, at reflux, afforded the carboxamides moderate to good yields, generally purified by recrystallization. In the second step, the benzanilides were converted to the corresponding thiobenzanilides with Lawesson’s reagent in toluene at 90 °C. Yields range from 13 to 91% and purification of the products always required column chromatography. The structure of all compounds was elucidated by extensive spectroscopic analysis (^1^H and ^13^C NMR, COSY, HMQC, HMBC, NOESY and FTIR) and by HR-ESI-MS spectrometry.

When studying the structure of aromatic amides and thioamides special attention must be given not only to the possible existence of (*E*) and (*Z*) rotamers along the NC(S) bond (Figure 4), but also to the presence of axial chirality if the arene and the thioamide moiety are not in the same plane [24], leading to the existence of atropoisomers. The restricted rotation about the Ar-CO(S), N-CO(S) and R-N bonds combined with their potential interaction, are responsible for the sometimes-complex conformational features of amides and their thioxo analogs [25]. The existence of atropoisomers must also be equated since chirality may profoundly affect biological activity.

The NMR structural characterization of the carboxamide precursors of the synthesized thiobenzanilides was straightforward and it was found that regardless of the substitution pattern or the substituents of the acyl group, only the presence of a set of well-defined signals was detected, indicating the occurrence in solution (either in CDCl_3_ or DMSO-*d*_6_) of just one rotamer. NOESY experiments revealed that the more stable (*Z*) rotamer was the only one present, in accordance with previous studies on secondary amides conformational equilibrium [26,27].

However, the NMR spectra of the thioamides disclosed that in solution their conformational equilibrium was dependent on the substitution of the acyl moiety and of the solvent used to run the experiment. The partially double-bond character of the C-N bond in thioamides is higher than in the corresponding amides due to the higher polarizability of the sulfur atom when compared to the oxygen, shaping the hindered rotation around the thioamide bond and the possibility of different conformational isomers, which can be observed in the NMR time scale. This fact was noticed mainly with compounds **1**–**5**, which have an *ortho*-nitrosubstitution. NMR spectra run in CDCl_3_ (either ^1^H or ^13^C) showed two sets of signals that uncovered mixtures of (*E*) and (*Z*) rotamers, with the (*Z*) one being just slightly predominant. Spectra were also run in DMSO-*d*_6_ having shown a marked decrease in the proportion of the minority rotamer, an indication that the conformational equilibrium was strongly dependent on the solvent used. Two-dimensional experiments enabled the separate assignment of all signals in both solvents.

The stereochemistry of both rotamers was disclosed by NOESY experiments run in DMSO-*d*_6_. The conformation of the major rotamer (Z) was deduced by the correlations of the NH proton with H-6 and H-2′/H-6′ protons. As expected, in the minor rotamer it was only possible to observe a cross-peak with the corresponding H-2′ and H-6′, and no correlation with H-6. The two rotamers of compound **1**, generated by PYMol, are presented in Figure 5a with the corresponding NOESY correlations.

Since dynamic NMR techniques have been largely used to study the restricted rotation around amide bonds, a variable-temperature ^1^H-NMR study was carried out to confirm the co-existence of the two rotamers on the NMR time scale. Figure 6 shows, as an example, the temperature-dependent spectra of compound **1** in DMSO-*d*_6_. Experiments were run at 25, 40, 55, 70, 85 and 115 °C. The distinctive and well-resolved NH, H-3 and CH_2_ signals for both rotamers were monitored, and it was found that the coalescence of the signals began to appear only at 70 °C, being completely achieved at 115 °C. When the sample was cooled down to 25 °C the signals of the two rotamers were restored, thus confirming that the duplication of all signals was due to a conformation equilibrium.

^1^H NMR spectra for compound **1** were also acquired in CDCl_3_, C_6_D_6_, THF-*d*_8_, (CD_3_)_2_CO and DMSO-*d*_6_. It was found that the rotational isomerism was dependent on the solvent, as referred above: for CDCl_3_ and C_6_D_6_ the ratio between the two rotamers was more balanced, while with hydrogen bond acceptor solvents the (*Z*) rotamer was clearly the predominant rotamer (Table 2).

The variation in the rotamer’s population in different solvents may be associated with factors such as Gibbs energy, solvation enthalpy or dipolar moments of the compounds. Solvents with high dipolar moments (e.g., DMSO-*d*_6_) increase the rotational barrier of the C(=S)-N bond, preventing the free rotation and thus favoring the more stable rotamer [28,29]. These results suggest that thioamides with a strongly polarized nitro group in the *ortho* position of the acyl moiety are more prone to be stabilized in the more stable rotamer in polar solvents. This is in accordance with results obtained by J. Kozic et al. [30]. Indeed, these authors found, for a 2-methoxybenzothioamide derivative, that the (*Z*)/(*E*) rotamer ratio ranged from 97:3 in (CD_3_)_2_CO to 86:14 in DMSO-*d*_6_, despite having not found any apparent relationship between the solvents’ properties and the (*Z*)/(*E*) conformational ratio.

For compounds **6**–**9** with an *ortho*-CF_3_ substitution or 15–18 with a *meta*-dinitrosubstitution, the ^1^H NMR spectra (taken in CDCl_3_) also showed two sets of signals that revealed mixtures of (*E*) and (*Z*) rotamers, with the (*Z*) one being strongly predominant, as proven by NOESY experiments. For these compounds, the minor rotamer, while detectable, could not always be quantified, nor with the signals assigned separately. In the case of compounds **10**–**14** with no substitution in the acyl group, the ^1^H NMR spectra only showed one set of signals, that were accurately assigned to the more stable (*Z*) conformer.

DFT calculations on both rotamers were also run for all 18 compounds in CDCl_3_ (Table 2), confirming that the (*Z*) conformer is consistently more stable than the (*E*) one (ΔΔ*G* values are always negative). For compound **1**, the solvent was also changed and the final (*Z*)/(*E*) ratio dependence on the solvent is in excellent agreement with the experimental data (Table 2).

To find out if the solid state of the predominant rotamer of compound **1** was also in the (*Z*) one, we carried out X-ray diffraction analysis. The X-ray crystal structure of compound **1** (ORTEP diagram) along with the atom numbering and the displacement ellipsoid plot is shown in Figure 5b. The structure revealed *quasi*-coplanarity of the thioamide and the *N*-aryl moiety, as evidenced by the S(1)-C(7)-N(2)-C(8) torsion angles of −1.3°. The 178° angle S(1)-C(7)-N(2)-H(2) disclosed the *trans* position of the NH and sulfur atoms, confirming that in the solid state, the (*Z*) rotamer also predominates.

### 2.2. Antiproliferative Activity

The cytotoxicity of thiobenzanilides 1–18 was evaluated in vitro in two human cancer cell lines, the A375 melanoma and the MCF-7 breast cancer cell lines, using the 3-(4,5-dimethylthiazol-2-yl)-2,5-diphenyl-tetrazolium bromide (MTT) assay. In this assay, the effect of different concentrations of thiobenzanilides to inhibit A375 and MCF-7 cell growth was analyzed after 24 h treatment. 

Dose–response curves were represented for each of the thiobenzanilides and parameters such as EC_50_, the lower and upper limit of the curve and the Hill slope were determined. The anticancer activity of the thiobenzanilides against A375 melanoma cells was seen to be concentration dependent and EC_50_ values in the range of µM concentrations were obtained (Table 3), except for compounds **13**, **14** and **16** which showed values higher than 100 µM. The chemotherapeutic agent doxorubicin was used as a positive control in this assay showing an EC_50_ value of 6.0 µM. As observed in Table 3 and more clearly visualized in Figure 7a, almost all the structural changes made on both rings of the parent thiobenzanilide led to increased cytotoxicity against this cell line, which suggests that ring substitution seems to be essential for cytotoxicity.

For the 2-NO_2_ series (**1**−**5** and **15**−**18**) the EC_50_ values are lower than those for the parent unsubstituted compounds, and again, the 4′- substitution seems to play a small influence in the cytotoxicity. The lowest value of EC_50_ (11.8 µM) for the A375 melanoma cells was obtained for compound **17**, which has a NO_2_ group at both R_2_ and R_3_ and an octyloxy group at R_4_ position. Compounds **1**, **15** and **18** presented the steepest dose–response curves with the highest Hill slope values (Table 3), meaning that a moderate dose increase significantly rises their antiproliferative activity, rendering these compounds likely more potent. However, thiobenzanilide **17** showed the lowest value (0.1 µM) for the threshold lower limit (Table 3), which represents the lowest dose at which cytotoxicity was detected. Therefore, compound **17** seems to be a promising candidate for future research. Dose–response curves for compound **17** and for the chemotherapeutic drug doxorubicin were compared and are shown in Figure 7b.

For the CF_3_ series (**6**−**9**), in particular for compounds **8** and **9**, substitution with a CF_3_ group at the R_1_ position increased the cytotoxicity of these thiobenzanilides, as compared to the corresponding unsubstituted compounds, compounds **14** and **11**, respectively. For these two compounds, low EC_50_ values, around 30 µM, were obtained with compound **8** showing also a low value for the threshold lower limit (0.5 µM). However, the remaining results suggest that within the CF_3_ series, other substitutions at the 4′ position were not determinant for the cytotoxicity, and in fact reduced it to some degree.

Structure–activity relationships were found to be intricated, since no straightforward correlation could be drawn between the structures of the tested compounds and their cytotoxicity against the A375 cell line.

Compounds **1**–**18** were also evaluated against the MCF-7 cell line, used for in vitro breast cancer studies, and compared with the positive control—the drug tamoxifen, clinically used for breast cancer therapy. An overall decrease in susceptibility to the compounds was observed, in a dose-dependent manner, when compared with the results for A375 cells. For most compounds, EC_50_ values could not be calculated since a high cell viability was observed for the higher tested concentration. Although not as evident as for the A375 cell line, results suggest that substitution in the aniline-derived moiety improves cytotoxicity. Again, the most meaningful result was obtained for one compound with R_2_ = R_3_ = NO_2_. Compound **15**, with a benzyl group at R_4_, showed the highest cytotoxicity with an EC_50_ value of 43 µM, which is not far away from tamoxifen’s EC_50_ value (30 µM).

### 2.3. ADME Predictions

The hit-to-lead phase is a critical phase of all drug-discovery programs, in which a first set of compounds with promising activities against a target is chemically changed into a lead molecule, after an iterative process of structure modification and activity improvement. The aim of this stage is to refine each hit series trying to produce more potent and selective compounds which possess adequate pharmacokinetic (PK) properties [31]. The experimental determination of PK parameters, an integral component of any small molecule discovery program, is a resource and time-consuming process, that evaluates properties such as absorption, distribution, metabolism and excretion (ADME). The use of in silico models to predict ADME properties allows researchers to save time and resources by immediately discarding compounds with unsuitable PK profiles before further optimization [32,33]. The knowledge of ADME properties is thus crucial to foresee the feasibility of a compound for therapeutic use [33].

In the present study, SwissADME [34] and pkCSM [35] chemoinformatic tools were used to predict the ADME properties of the synthesized compounds **1**–**18**. These results are available in the Appendix A. Properties such as logarithm of partition coefficient (log P), molecular weight (MW), TPSA (topological polar surface area), number of hydrogen bond acceptors and hydrogen bond donors and rotatable bonds, were calculated. ADME prediction, focusing on gastrointestinal (GI) absorption and Caco 2 permeability, blood–brain barrier (BBB) permeability, P-glycoprotein substrate and other absorption properties, cytochromes P450 (CYP) metabolism and other parameters of excretion and toxicity, were also evaluated. Additionally, drug-likeness descriptors such as Lipinski’s rule of five and bioavailability score, among others, were computed as well. These results were compared with the predictions for other thiobenzanilides reported as also having an anticancer effect on cell lines A375 (Figure 2) [21] and MCF7 (Figure 3) [22].

Based on the results, it is worth noticing that, overall, compounds (**1**–**18**) show adequate molecular properties, with the majority of them meeting the criteria established by Lipinski’s rule of five, showing no violations or one at maximum (except for compound 16 with two violations in one of the criteria), and with no major differences when compared with the previous reported thiobenzanilides. As for the absorption parameters, the majority of the compounds and the previously reported thiobenzanilides are predicted to exhibit high GI permeability and not to be substrates for P-glycoprotein efflux, neither to have good BBB permeability. The predicted steady-state volume of distribution (VDss) values for most compounds vary between −0.15 and 0.45 [33] which indicates that they have a steady distribution in blood plasma and tissues. All compounds are substrates for CYP3A4, a measure of easiness to be metabolized and most of them are not hepatotoxic.

For hit compounds **8**, **9** and **17**, showing the highest potency against melanoma A375 cancer cells, compound **8** shows the best ADME results. In the case of compounds **9** and **17**, the exhibited pharmacokinetic profile needs to be refined mostly to improve GI permeation and prevent hepatotoxicity. The same type of improvements must be addressed for compound **15**, which was shown to have the most promising experimental results of antiproliferative activity against breast cancer MCF7 cells.

## 3. Materials and Methods

### 3.1. Chemistry

#### 3.1.1. General Methods

All solvents and reagents were obtained from commercial suppliers with an analytical grade and were used without further purification, except for pyridine and toluene which were refluxed and distilled using standard purification procedures.

Analytical thin layer chromatographies (TLC) were performed on silica gel 60 F_254_ (Merck^®®^ Ref. 5554), neutral alumina oxide F_254_ neutral (Merck^®®^ Ref. 5581) or silica gel 60 RP-18 F_254_ (Merck^®®^ Ref. 5559) plates; preparative thin layer chromatographies (PTLC) were run on silica gel 60 F_254_ plates (Machery-Nagel Ref. mv 809053); spots were visualized with UV light (254 nm). Column chromatographies were performed on silica-gel 60 (0.004–0.066 mm) (Merck^®®^ Ref. 9385 or Scharlau GE00481000), aluminium oxide 90 active neutral 0.063–0.200 mm (Merck^®®^ Ref. 1077) or reverse phase silica-gel 60 RP-18 (Merck^®®^ Ref. 711021.1000).

NMR spectra were recorded on a Bruker Avance Ultra Shield^TM^ spectrometer, operating at 400.13 MHz for ^1^H and at 100.60 MHz for ^13^C; chemical shifts were expressed as δ values and referenced to the residual solvent peak (DMSO-*d_6_*: δ_H_ = 2.50 ppm, δ_C_ = 39.5 ppm; CDCl_3_: δ_H_ = 7.26 ppm, δ_C_ = 77.0 ppm; CD_3_OD: δ_H_ = 3.35 ppm, δ_C_ = 49.9 ppm; C_6_D_6_: δ_H_ = 7.16 ppm, δ_C_ = 128.1 ppm; (CD_3_)_2_CO: δ_H_ = 2.05 ppm, δ_C_ = 206.0, 29.8 ppm; THF-*d*_8_: δ_H_ = 1.72 and 3.58 ppm, δ_C_ = 39.5 ppm); coupling constants were reported in units of Hertz (Hz). FTIR spectra were obtained using a Nicolet 6700 (Thermo Electron Corp., Madison, WI, USA) in KBr pellets and only the diagnostic absorption bands were reported in cm^−1^.

Melting points were taken in open capillary tubes using an apparatus Stuart^®®^ SMP 30 or in a Reichert Thermovar Microscope and are uncorrected.

HR-ESI-MS spectra were acquired in the positive or negative mode on a hybrid quadrupole time-of-flight (QTOF) Bruker Impact II mass spectrometer and the data were acquired through the Data Analysis 4.4 software. Samples were solubilized in dichloromethane and the following parameters were used: end plate offset: −500 V; capillary voltage: +3.5 kV; nebulizer: 0.3 Bar; dry gas: 4 L/min; heater temperature: 200 °C; *m*/*z*: 50–1500.

The X-ray single crystal data of 1 were collected with monochromated Mo-Kα radiation (λ = 0.71073 Å) on a Bruker SMART Apex II diffractometer equipped with a CCD area detector at 150 (2) K. Data reduction was carried out using the SAINT-NT software package and multi-scan absorption correction was applied using the SADABS program. The structures were solved by a combination of direct methods and subsequent different Fourier syntheses followed by successive refinements by full-matrix least squares on *F*2 using the SHELX-2013 suite. The hydrogen atoms were inserted at ideal geometric positions and refined with Uiso = 1.2 Ueq of the parent carbon atom. Molecular and crystal packing diagrams were drawn with the Mercury software package.

#### 3.1.2. General Procedure for the Synthesis of the Thiobenzanilide Derivatives **1**–**18**

Thiobenzanilides **1**–**18** were obtained from the parent carboxamides prepared by a Schotten-Baumann reaction. Briefly, one equivalent of the acid chloride was added dropwise to a mixture of the amine dissolved in pyridine with stirring and was cooled in an ice bath. The reaction mixture was then refluxed until completion of the reaction, cooled to room temperature, and thereafter, poured into water, and extracted with dichloromethane and ethyl acetate. The combined organic extracts were washed with NaHCO_3_ 5%, dried over anhydrous Na_2_SO_4_ and the solvent evaporated under a vacuum. Crystallin solids were obtained after recrystallization with the appropriate solvent.

Then, to a stirred warm solution of the benzanilide derivatives synthesized as previously described (5 mmol) in anhydrous toluene (10 mL) was added Lawesson’s reagent (10 mmol), portion-wise. The reaction mixture was heated at 90 °C and monitored by TLC until completion. The toluene was removed under vacuum and the so-obtained residue was purified by column chromatography and recrystallization.

*N*-(4-benzylphenyl)-2-nitrobenzothioamide (**1**)

Yellow needles (purification by column chromatography over neutral alumina oxide with hexane/AcOEt 75:25, followed by recrystallization with ethyl ether); yield: 58%. IR (KBr): ν_max_/cm^−1^: 3218 (N-H st), 3183-2861 (C-H aromatic st), 1601 (C-C aromatic st), 1528 (NO_2_ st), 1351 (C-NO_2_ st), 1179 (C=S st). ^1^H-NMR [DMSO-d_6,_ 400.13 MHz]: 12.25 (s, NH), 8.10 (dd, 1H, *J = *8.2. 0.9 Hz, H3), 7.80 (m, 1H, H5), 7.79 (d, 2H, *J = *8.2 Hz, H2′/H6′), 7.65 (td, 1H, *J = *8.0, 1.4 Hz, H4), 7.61 (dd, 1H, *J = *7.6 Hz, 1.2, H6), 7.32 (d, 2H. *J = *8.5 Hz, H3′/H5′), 7.29 (d, 2H, *J = *7.1 Hz, H3″/H5″), 7.26 (m. 2H. H2″/H6″), 7.19 (tt. 1H. *J = *7.1. 1.5 Hz. H4″), 3.96 (s. 2H. CH_2_ ). ^13^C-NMR [DMSO-d_6_, 100.60 MHz]: 193.23 (C=S), 145.08 (C2), 141.13 (C1″), 139.80 (C4′), 139.09 (C1), 137.30 (C1′), 133.93 (C5), 129.80 (C4), 129.08 (C6), 128.96 (C3′/C5′), 128.71 (C3″/C5″), 128.52 (C2″/C6″), 126.08 (C4″), 124.33 (C3), 123.13 (C2′/C6′), 40.71 (CH_2_). HR-ESI-MS *m*/*z* 349.1002 [M+H]^+^ (calcd for C_20_H_16_N_2_O_2_SH. 349.1005).

2-nitro-*N*-(4-phenoxyphenyl)benzothioamide (**2**)

Yellow amorphous solid (purification by two subsequent column chromatographies over silica-gel 60 (0.004-0.066 mm) with hexane/AcOEt 65:35); yield: 45%. IR (KBr): ν_max_/cm^−1^: 3036 (C-H aromatic st), 1588 (C-C aromatic st), 1527 (NO_2_ st), 1342 (C-NO_2_ st), 1242 (C=S st), 1020 (C-OC st). ^1^H-NMR [DMSO-*d*_6_, 400.13 MHz]: 10.40 (s, NH); 8.20 (d, 1H, 8.2 Hz, H3); 8.00 (d, 2H. 8.9 Hz, H2′-H6′); 7.91 (t, 1H, 7.5 Hz, H5); 7.77 (m, 1H, H4); 7.74 (m, 1H, H6); 7.52 (t, 2H, 7.6 Hz, H3″/H5″); 7.27 (d,1H, 7.4 Hz, H4″); 7.22 (d, 2H, 8.5 Hz, H3′/H5′); 7.16 (d, 2H, 7.8 Hz, H2″/H6″). ^13^C-NMR [DMSO-*d*_6_, 100.60 MHz]: 193.16 (C=S); 156.54 (C1″); 154.62 (C4′); 145.09 (C2); 139.02 (C1); 134.72 (C1′); 133.95 (C5); 130.16 (C3″/C5″); 129.84 (C4); 129.10 (C6); 124.82 (C2′/C6′); 124.36 (C3); 123.78 (C4″); 118.71 (C3′/C5′); 118.69 (C2″/C6″). HR-ESI-MS *m*/*z* 351.0795 [M+H]^+^ (calcd for C_19_H_14_N_2_O_3_S +H^+^, 351.0798).

2-nitro-*N*-(4-octylphenyl)benzothioamide (**3**)

Yellow amorphous solid (purification by two subsequent column chromatographies over silica-gel 60 (0.004–0.066 mm) with hexane/AcOEt 74:26 and hexane/ethyl ether 53:47, followed by column chromatography over reverse-phase silica-gel 60 RP-18 with methanol); yield: 13%. IR (KBr): ν_max_/cm^−1^: 3110 (aliphatic C-H st), 2920–2849 (aromatic CH st), 1527 (C-C aromatic st),1508 (NO_2_ st), 1382 (C-NO_2_ st), 1239 (C=S st). ^1^H-NMR [DMSO-*d*_6_, 400.13 MHz]: 12.23 (s. NH); 8.10 (d, 1H, *J =* 8.1 Hz, 1.0 Hz. H3); 7.81 (td. 1H. *J =* 7.5. 1.2 Hz. H5); 7.77 (d. 2H. *J =* 8.5 Hz, H2′/H6′); 7.66 (m. 1H. H4); 7.62 (dd, 1H, *J =* 7.7, 1.2 Hz, H6); 7.27 (d. 2H, *J =* 8.4 Hz, H3′/H5′); 2.59 (t. 2H. *J =* 7.5 Hz, H1″); 1.57 (m. 2H. H2″); 1.26 (m, 10H, H3″-H7″). 0.85 (t. 3H. *J* = 6.8 Hz H8″). ^13^C-NMR [DMSO-*d*_6_, 100.60 MHz]: 193.03 (C=S); 145.10 (C2); 140.90 (C4′); 139.12 (C1); 137.00 (C1′); 133.92 (C5); 129.77 (C4); 129.08 (C6); 128.49 (C3′/C5′); 124.33 (C3); 122.89 (C2′/C6′); 34.79 (C1″). 31.33 (C6″); 30.97 (C2″); 28.87 (C3″); 28.71 (C4″); 28.67 (C5″); 22.12 (C7″); 14.00 (C8″). HR-ESI-MS *m*/*z* 371.1788 [M+H]^+^ (calcd for C_21_H_26_N_2_O_2_S+H^+^. 371.1788).

2-nitro-*N*-[4-octyloxy)phenyl]benzothioamide (**4**)

Yellow oil (purification by column chromatography over silica-gel 60 (0.004–0.066 mm) with hexane/AcOEt 69:31, followed by preparative thin-layer chromatography with the same eluent); yield: 15%. IR (KBr): ν_max_/cm^−1^: 3115 (aliphatic C-H st). 2921-2855 (aromatic CH st), 1573 (C-C aromatic st), 1528 (NO_2_ st), 1388 (C-NO_2_ st), 1245(C=S st). ^1^H-NMR [DMSO-*d*_6_, 400.13 MHz]: 12.15 (s, NH); 8.09 (d, 1H, 8.2 Hz, H3); 7.80 (m, 1H, H5); 7.77 (m, 2H, H2′/H6′); 7.66 (m, 1H, H4); 7.62 (m, 1H, H6); 6.99 (d, 2H, 9.0 Hz, H3′/H5′); 3.98 (t. 2H, 6.5 Hz, H1″); 1.72 (quint, 2H, 6.9 Hz, H2″); 1.41 (m, 2H, H3″); 1.30-1.27 (m, 8H, H4″-H7″); 0.86 (m, 3H, H8″). ^13^C-NMR [DMSO-*d*_6_, 100.60 MHz]: 192.47 (C=S); 156.89 (C4′); 145.14 (C2); 139.06 (C1); 133.85 (C5); 132.17 (C1′); 129.70 (C4); 129.06 (C6); 124.50 (C2′/C6′); 124.28 (C3); 114.28 (C3′/C5′); 67.66 (C1″); 31.26 (C6″); 28.76 (C5″); 28.70 (C4″); 28.66 (C2″); 25.54 (C3″); 22.11 (C7″); 13.98 (C8″). HR-ESI-MS *m*/*z* 387.1732 [M+H]^+^ (calcd for C_21_H_26_N_2_O_3_S+H^+^, 387.1737).

2-nitro-*N*-phenylbenzothioamide (**5**)

Yellow amorphous solid (purification by column chromatography over silica-gel 60 (0.004–0.066 mm) with hexane/ethyl ether 40:60); yield: 15%. IR (KBr): ν_max_/cm^−1^: 3208 (N-H st), 2955-2894 (C-H aromatic st), 1591 (C-C aromatic st). 1525 (NO_2_ st), 1336 (C-NO_2_ st), 1170 (C=S st). ^1^H-NMR [DMSO-*d*_6_, 400.13 MHz]: 12.30 (s, NH); 8.10 (d, 1H, 8.2 Hz, H3); 7.88 (d, 2H, 7.6 Hz, H2′/H6′); 7.47 (t, 1H, 7.8 Hz, H3′/H5′); 7.68 (m, 1H, H4); 7.65 (m, 1H, H6); 7.47 (m, 2H, H3′/H5′); 7.31 (m, 1H, H4′ ); 7.88 (m, 2H, H2′/H6′). ^13^C-NMR [DMSO-*d_6_*, 100.60 MHz]: 193.60 (C=S); 145.06 (C2); 139.23 (C1); 139.08 (C1′); 133.94 (C5); 129.81 (C4); 129.08 (C6); 128.76 (C3′/C5′); 126.60 (C4′); 124.33 (C3); 123.03 (C2′/C6′). HR-ESI-MS *m*/*z* 259.0529 [M+H]^+^ (calcd for C_13_H_10_N_2_O_2_S+H^+^, 259.0536).

*N*-(4-octylphenyl)-2-(trifluoromethyl)benzothioamide (**6**)

Yellow amorphous solid (purification by two subsequent column chromatographies over silica-gel 60 (0.004–0.066 mm) with hexane/AcOEt 85:15 and hexane/AcOEt 70:30); yield: 37%. IR (KBr): ν_max_/cm^−1^: 3170 (N-H st), 3108-2854 (C-H aromatic st), 1509 (C-C aromatic st), 1312 (CF st). 1125 (C=S st). 770 (CF_3_ st). ^1^H-NMR [CDCl_3_, 400.13 MHz]: 8.76 (s. NH); 7.70 (m, 2H, H3,H6); 7.64 (m, 2H, H2′/H6′); 7.61 (d, 1H, H5); 7.51 (t, 1H, 7.8 Hz, H4); 7.26 (m, 2H, H3′/H5′); 2.63 (t, 2H, 7.6 Hz, H1′); 1.62 (quint, 2H, 7.3 Hz, H2″); 1.29 (m,10H. H3″-H7″); 0.87 (m, 3H, H8″). ^13^C-NMR [CDCl_3_, 100.60 MHz]: 196.62 (C=S); 142.75 (C1); 142.47 (C4′); 135.88 (C1′); 132.17 (C5); 130.03 (C6); 129.27 (C4); 129.02 (C3′/C5′); 126.25 (q. ^3^J_CF_ = 5. C3); 124.96 (q. ^2^J_CF_ = 32.0 C2); 125.85 (q. ^1^J_CF_ = 275. CF_3_); 123.41 (C2′/C6′); 35.63 (C1″); 31.87 (C6″); 31.34 (C2″); 29.45 (C5″); 29.30 (C4″); 29.27 (C3″); 22.67 (C7″); 14.10 (C8″). HR-ESI-MS *m*/*z* 392.1659 [M-H]^+^ (calcd for C_22_H_26_F_3_NS-H^+^, 392.1665).

*N*-(4-(octyloxi)phenyl)-2-(trifluoromethyl)benzothioamide (**7**)

Yellow amorphous solid (purification by column chromatography over silica-gel 60 (0.004–0.066 mm) with hexane/AcOEt 80:20); yield: 86%. IR (KBr): ν_max_/cm^−1^: 3233 (N-H st), 3065-3040 (C-H aromatic st), 1510 (C-C aromatic st), 1249 (C=S st), 767 (CF_3_ st). ^1^H-NMR [CDCl_3_, 400.13 MHz]: 8.78 (s, NH); 7.70 (m, 2H, H3; H6); 7.60 (m, 3H. H5; H2′/H6′); 7.51 (t, 1H, 7.7 Hz, H4); 6.95 (d, 2H, 8.5 Hz, H3′/H5′); 3.97 (t, 2H, 6.4 Hz, H1′); 1.78 (quint, 2H, 6.9 Hz, H2″); 1.46 (m, 2H, H3″); 1.31 (m, 8H, H4″-H7″); 0.89 (m, 3H, H8″). ^13^C-NMR [CDCl_3_,100.60 MHz]: 196.78 (C=S); 158.13 (C4′); 142.65 (C1); 131.00 (C1′); 132.14(C5); 130.03 (C6); 129.27 (C4); 126.22 (q. ^3^J_CF_ = 5. C3); 125.04 (C2′/C6′); 114.76 (C3′/C5′); 68.25(C1″); 31.78(C6″); 29.31. 29.22. 29.17 (C2″; C4″; C5′); 26.00(C3″); 22.64 (C7″); 14.09 (C8″). HR-ESI-MS *m*/*z* 408.1614 [M-H]^+^ (calcd for C_22_H_26_F_3_NOS-H^+^, 408.1614).

*N*-phenyl-2-(trifluoromethyl)benzothioamide (**8**)

Yellow oil (purification by preparative thin column chromatography using hexane/AcOEt 80:20); yield: 21%. IR (KBr): ν_max_/cm^−1^: 3107 (N-H st), 2994-2885 (C-H aromatic st), 1594 (C-C aromatic st), 1313 (CF st), 1175 (C=S st), 770 (CF_3_ st). ^1^H-NMR [CDCl_3_, 400.13 MHz]: 8.80 (s, NH); 7.76 (d, 2H, 7.9 Hz, H2′/H6′); 7.70 (m, 2H, H3; H6); 7.61 (t, 1H, H5); 7.52 (t, 1H, 7.6 Hz, H4); 7.46 (t, 2H, 7.6 Hz, H3′/H5′); 7.33 (t, 1H, H4′). ^13^C-NMR [CDCl_3_,100.60 MHz]: 196.98 (C=S); 142.74 (C1); 138.24 (C1′); 132.20 (C5); 130.01 (C6); 129.34 (C4); 129.19 (C3′/C5′); 127.41 (C4′); 126.27 (q. ^3^J_CF_ = 5. C3); 124.94 (q. ^2^J_CF_ = 32.0 C2); 123.83 (q. ^1^J_CF_ = 275. CF_3_); 123.56 (C2/C6′). HR-ESI-MS *m*/*z* 282.0545 [M+H]^+^ (calcd for C_14_H_10_F_3_NS+H^+^, 282.0559).

*N*-(4-phenoxyphenyl)-2-(trifluoromethyl)benzothioamide (**9**)

Yellow crystals (purification by column chromatography over silica-gel 60 (0.004–0.066 mm) with hexane/AcOEt 84:16); followed by recrystallization with ethanol); yield: 82%. IR (KBr): ν_max_/cm^−1^: 3163 (N-H st), 2997-2901(C-H aromatic st), 1588 (C-C aromatic st), 1319 (CF st). 1243 (C=S st). ^1^H-NMR [CDCl_3_, 400.13 MHz]: 8.78 (s, 1H. N-H); 7.70 (m, 4H, H3; H6; H2′/H6′); 7.61 (t, 1H, *J* = 7.4 Hz, H5); 7.50 (t, 1H. *J* = 7.8 Hz, H4); 7.37 (m, 2H, *J* = 8.0 Hz, H3”/H5”); 7.15 (t, 1H, *J* = 7.4 Hz, H4”); 7.06 (4H, m, H3′; H5′; H2”/H6”). ^13^C-NMR (CDCl_3,_ 100.6 MHz): 196.89 (C=S); 156.51 (C1”); 156.29 (C4′); 142.56 (C1); 133.18 (C1′); 132.19 (C5); 130.1 (C6); 129.9 (C3”/C5”); 129.4 (C4); 126.32 (q. ^3^J_CF_ = 5, C3); 125.4 (C2′/C6′);123.87 (C-4”); 119.39 (C3′/C-5′); 118.84 (C2”/C-6”). HR-ESI-MS *m*/*z* 372.0675 [M-H]^+^ (calcd for C_20_H_14_F_3_NOS-H^+^, 372.0675).

*N*-(4-benzylphenyl)benzothioamide (**10**)

Yellow crystals (purification by recrystallization with ethanol); yield: 69%. M.p.: 149.8–150.3 °C. IR (KBr): ν_max_/cm^−1^: 3179 (N-H st), 2923 (C-H aromatic st), 1637 (C-C aromatic st), 1508 (C-N st), 1213 (C=S st). ^1^H-NMR [CDCl_3_,400.13 MHz]: 9.00 (s, 1H, NH); 7.86 (d, 2H, *J =* 7.4 Hz, H2/H6); 7.70 (d, 2H, *J =* 8.2 Hz; H-2′/H6′); 7.53 (t, 1H, *J =* 7.2 Hz, H4); 7.46 (t, 2H, H-3/H5); 7.33 (m, 1H, H4″), 7.26 (m, 4H, H3′/H5′, H2″/H6″); 7.24 (d, 2H, H3″/H5″), 4.04 (s, 2H, CH_2_); ^13^C-NMR (CDCl_3_, 100.6 MHz,): 198.2 (C=S), 143.2 (C1), 140.6 (C1″), 140.2 (C4′), 137.1 (C1′), 131.3 (4), 129.5 (C-3′/C5′), 129.0 (C2″/C6″), 128.7 (C3/C5), 128.6 (C3″/C5″), 126.7 (C2/C6), 126.3 (C4″), 123.8 (C2′/C6′), 41.6 (CH_2_). HR-ESI-MS *m*/*z* 302.1007[M-H]^+^ (calcd for C_20_H_17_NS-H^+^, 302.1009).

*N*-(4-phenoxyphenyl)benzothioamide (**11**)

Yellow crystals (purification by column chromatography over silica-gel 60 (0.004–0.066 mm) with hexane/AcOEt 81:19.5); yield: 56%. IR (KBr): ν_max_/cm^−1^: 3176 (N-H st), 3046 (C-H aromatic st), 1587 (C-C aromatic st), 1488 (C-N st). 1166 (C=S st). ^1^H-NMR [CDCl_3_, 400.13 MHz]: 9.25 (s, 1H, NH); 7,85 (d, 2H, *J =* 7.3 Hz, H2/H6); 7.72 (d, 2H, *J =* 8.8 Hz; H2′/H6′); 7.48 (t, 1H, *J =* 7.3 Hz, H4); 7.42 (t, 2H, *J =* 7.4 Hz, H3/H5); 7.35 (t, 2H, *J =* 7.9 Hz, H3″/H5″), 7.13 (t, 1H, *J =* 8.8 Hz; H4″); 7.05 (m, 4H, H3′/H5′, H2″/H6″); ^13^C-RMN (CDCl_3_,100.6 MHz,): 198.31 (C=S); 156.59 (C1″); 155.80 (C4′); 142.94 (C1); 134.11 (C1′); 131.18 C4); 129.82 (C-3″/C-5″); 128.54 (C3/C5); 126.72(C2/C6); 125.42 (C2′/C6′); 123.69 (C4″); 119.22 (C-2″/C-6″) 118.68 (C-3′/C-5′). HR-ESI-MS *m*/*z* 306.0940 [M+H]^+^ (calcd for C_19_H_15_NOS+H^+^, 306.0947).

*N*-(4-octylphenyl)benzothioamide (**12**)

Yellow amorphous solid (purification by two subsequent column chromatographies over silica-gel 60 (0.004–0.066 mm) with hexane/AcOEt 74:26 and hexane/ethyl ether 53:47, followed by reverse-phase column chromatography with methanol); yield: 42%. IR (KBr): ν_max_/cm^−1^: 3110 (N-H st), 2920-2849 (C-H aromatic st), 1527 (C-C aromatic st), 1508 (NO_2_ st). 1239 (C=S st). ^1^H-NMR [CDCl_3_, 400.13 MHz]: 8.93 (s. NH); 7.87 (d, 1H, *J =* 7.4 Hz, H3); 7.69 (m, 2H, *J =* 8,1 Hz, H2′/H6′); 7.53 (t, 1H, H4); 7.46 (t, 1H, *J =* 6.9 Hz, H3/H5); 7.25 (m, 2H, H3′/H5′); 2.63 (t, 2H, 7.6 Hz, H1″); 1.63 (m, 2H, H2″); 1.27 (m, 10H, H3″-H7″); 0.87 (m, 3H, H8″). ^13^C-NMR [CDCl_3_,100.60 MHz]: 194.96 (C=S); 145.30 (C2); 142.56 (C4′); 139.39 (C1); 135.70 (C1′); 133.72 (C5); 129.75 (C4); 129.05 (C3′/C5′); 128.98 (C6); 124.74 (C3); 123.59 (C2′/C6′); 35.62 (C1″); 31.86 (C2″); 31,33 (C6″); 29.44 (C5″); 29.29 (C4″); 29.23 (C3″); 22.65 (C7″); 14.10 (C8″). HR-ESI-MS *m*/*z* 371.1788 [M+H]^+^ (calcd for C_21_H_26_N_2_O_2_S+H^+^, 371.1788).

*N*-(4-(octyloxy)phenyl)benzothioamide (**13**)

Yellow crystals (purification by column chromatography over silica-gel 60 (0.004–0.066 mm) with hexane/AcOEt 85:15; yield: 14%. M.p.: 100–101 °C. IR (KBr): ν_max_/cm^−1^: 3154 (NH st), 3045-3005 (aromatic CH st), 1213(C=S st). ^1^H-NMR [CDCl_3_,, 400.13 MHz]: 8.95 (s, NH); 7.83 (d, 7.6 Hz, H2/H6), 7.60 (d, 2H, 8.5 Hz, H2′/H6′); 7.49 (m, 1H, H4); 7.42 (t, 3H, H3/H5); 6.93 (d, 2H, 8.5 Hz, H3′/H5′); 3.97 (t. 2H, 12.8 Hz, H1″); 1.79 (m, 2H, H2″); 1.45 (m, 2H, H3″); 1.31 (m, 8H, H4″-H7″); 0.89 (m, 3H, H8″). ^13^C-NMR [CDCl_3_, 100.60 MHz]: 198.20 (C=S); 157.81 (C4′); 142.87 (C1); 131.73 (C1′); 131.15 (C4); 128.56 (C3/C5); 126.67 (C2/C6); 125.52 (C2′/C6′); 114.63 (C3′/C5′); 68.22 (C1″); 31.77 (C6″); 29.30, 29.20, 29.17 (C2″; C4″; C5′); 25.99 (C3′); 22.62 (C7″); 14.07 (C8″). HR-ESI-MS *m*/*z* [M-H]^+^ 340.1740 (calcd for C_21_H_27_NOS-H^+^, 340.1741).

*N*-phenylbenzothioamide (**14**)

Yellow crystals (purification by recrystallization with ethanol); yield: 72%. M.p.: 100.0–101.8 °C). IR (KBr): ν_max_/cm^−1^: 3161 (NH st), 3023-2960 (aromatic CH st), 1445 (C-C aromatic st), 1206 (C=S st). ^1^H-NMR [CDCl_3_, 400.13 MHz]: 9.09 (s, NH); 7.86 (d, 2H, 6.7 Hz, H2/H′6); 7.77 (d, 2H, 7.0 Hz H2′/H6′); 7.48 (m, 1H, H4); 7.43 (m, 4H, H3/H5, H3′/H5′)); 7.30 (t, 1H, H4′). ^13^C-NMR [CDCl_3_,100.60 MHz]: 198.40 (C=S); 143.02 (C1); 138.90 (C1′); 131.25 (C4); 128.98 (C3′/C5′); 128.58 (C3/C5); 126.95 (C2/C6); 126.65 (C4′); 123.6 (C2′/C6′). HR-ESI-MS *m*/*z* 212.0540 [M-H]^+^ (calcd for C_13_H_11_NS-H^+^, 212.0539).

*N*-(4-benzylphenyl)-3,5-dinitrobenzothioamide (**15**)

Yellow crystals (purification by column chromatography over silica-gel 60 (0.004–0.066 mm) with toluene/ethyl ether 92:8, followed by recrystallization with methanol*)*; yield: 91%. ^1^H-NMR [CDCl_3_, 400.13 MHz]: 9.24 (s, 1H, NH); 9.13 (s, 1H, H4); 8.98 (s, 2H, H2/H6); 7.69 (d, 2H, *J = *7.8 Hz, H2′/H6′); 7.32 (m, 2H, H3′/H5′); 7.22 (t, 2H, H3″/H5″); 4.03 (m. 2H. CH_2_).^13^C-NMR [CDCl_3_, 100.60 MHz]: 191.42 (C=S); 148.36 (C3/C5); 141.22 (C4′); 140.25 (C1″); 136.28 (C1′); 129.77 (C3′/C5′); 129.00 (C2″/C6″); 128.66 (C3″/C5″); 126.67 (C2/C6); 126.42 (C4″); 123.59 (C2′/C6′), 120.28 (C4); 41.56 (CH_2_). HR-ESI-MS *m*/*z* 360.1003 [M-H]^+^ (calcd for C_20_H_15_N_3_O_4_-H^+^, 360.0990).

*N*-(4-octylphenyl)-3,5-dinitrobenzothioamide (**16**)

Yellow crystals (purification by column chromatography over silica-gel 60 (0.004–0.066 mm) with toluene/ethyl acetate 98:2, followed by recrystallization with methanol*)*; yield: 74%. ^1^H-NMR [CDCl_3,_ 400.13 MHz]: 9.22 (s, 1H, NH); 9.13 (s, 1H, H4); 8.99 (s, 2H, H2/H6); 7.67 (d, 2H*, J =* 8.4 Hz, H2′/H6′); 7.32 (m, 2H, *J =* 82 Hz, H3′/H5′); 2.65 (t, 2H, *J =* 7.7 Hz, H1″); 1.68 (m, 2H, H2″); 1.29 (m, 10H, H3″-H7″); 0.88 (m, 3H, H8″). ^13^C-NMR [CDCl_3_, 100.60 MHz]: 191.6(C=S); 148.36 (C3/C5); 143.4 (C4′); 135.70 (C1′); 129.3 (C3′/C5′); 126.7 (C2/C6); 123.4 (C2′/C6′); 120.1 (C4); 35.7 (C1″); 31.8 (C6″); 31.3 (C2″); 29.5, 29.3, 29.2 (C3″-C5″); 22.7 (C7″); 14.10 (C8″). HR-ESI-MS *m*/*z* 414.1491 [M-H]^+^ (calcd for C_21_H_25_N_3_O_4_S-H^+^, 414.1493).

3,5-dinitro-*N*-(4-(octyloxy)phenyl)benzothioamide (**17**)

Yellow crystals (purification by column chromatography over silica-gel 60 (0.004–0.066 mm) with toluene/ethyl ether 98:2, followed by recrystallization with methanol*)*; yield: 56%. M.p.: 98.5–100.4 °C. IR (KBr): ν_max_/cm^−1^: 3248 (NH st), 3088-3055 (aromatic CH st), 2922-2854 (aliphatic CH st), 1597 (C-C aromatic st), 1537 (NO_2_ st), 1388 (C-NO_2_ st), 1247 (C=S st). ^1^H-NMR [acetone-*d*_6_, 400.13 MHz]: 11.57 (s, 1H, NH); 9,10 (s, 2H, H2/H6); 9.03 (s, 1H, H4); 7.84 (t, 2H, H2′/H6′); 7.04 (d, 2H, *J* = 9.0, H3′/H5′); 4.05 (t, 2H, *J* = 6.5, H1″); 1.80 (m, 2H, H2″); 1.50 (m, 2H, H3″); 1.31 (m, 8H, H4″-H7″); 0.89 (t, 3H, H8″). ^13^C-NMR [acetone-*d_6_,* 100.60 MHz]: 191.99 (C=S); 158.79 (C4′); 148.90 (C3/C5); 133.20 (C1′); 127.99 (C2/C6); 126.02 (C2′/C6′); 120.39 (C4); 114.49 (C3′/C5′); 68,67 (C1″); 32.56 (C6′); 30.07 (C2″, C4″, C5″); 26.62 (C3″); 23.17 (C7″), 14.21 (C8′). HR-ESI-MS *m*/*z m*/*z* 430.1445 [M-H]^+^ (calcd for C_21_H_25_N_3_O_5_S-H^+^, 430.1442).

3,5-dinitro-*N*-phenylbenzothioamide (**18**)

Yellow crystals (purification by column chromatography over silica-gel 60 (0.004–0.066 mm) with toluene/ethyl ether 95:5, followed by recrystallization with methanol*)*; yield: 34%. IR (KBr): ν_max_/cm^−1^: 3248 (NH st), 3089-2872 (aromatic CH st), 1620 (C-C aromatic st), 1598 (NO_2_ st), 1388 (C-NO_2_ st), 1344 (C=S st). ^1^H-NMR [acetone-*d_6_*, 400.13 MHz]: 9.11 (s, 2H, H2/H6); 9.03 (s, 1H, H4); 7.96 (d, 2H, *J* = 8.3, H2′/H6′); 7.49 (t, 2H, *J* = 7.9, H3′/H5), 7.35 (t, 1H, *J* = 7.43, H4′). ^13^C-NMR [acetone-*d_6_*, 100.60 MHz]:192.99 (C=S); 148.97, 148.94 (C3); 140.41 (C1′); 129.67 (C3′/C5′); 128.38 (C2/C6); 127.92 (C4′); 124.69 (C2′/C6′); 120.68 (C4). HR-ESI-MS m/z 302.0238 [M-H]^+^ (calcd for C_13_H_9_N_3_O_4_S-H^+^, 302.0241).

#### 3.1.3. Computational Methods

The initial structures for all compounds were built using Chemcraft [36]. All *Z* conformers were converted in the *E* configuration but rotating 180 degrees in the respective C–N bond. The QM calculations were performed with Gaussian [37] at the density functional theory (DFT) level with the B3LYP functional and 6–31G* basis set [38]. The different solvent calculations used the SMD implicit solvent model [39]. The nature of the stationary structures was confirmed by frequency calculations. The ΔΔ*G* values are the differences in the Gibbs energies of the two conformers and were obtained according to ΔΔ*G* = Δ*G*(*Z*) − Δ*G*(*E*). The (*E*) population fractions were calculated from the equation f(*E*) = e^(ΔΔ^*^G^*^/RT)^.

#### 3.1.4. Growth Inhibition Activity Assay against Human Cancer Cell Lines

MCF-7 human mammary gland adenocarcinoma epithelial and A375 human skin malignant melanoma cell lines were purchased from American Type Culture Collection (ATCC, Manassas, VA, USA). MCF-7 cells (ATCC#HTB-37) and A375 cells (ATCC#CRL-1619) were cultured in Dulbecco’s Modified Eagle’s Medium (DMEM) supplemented with 10% foetal bovine serum, 100 U/mL penicillin-streptomycin, and 2 mM L-glutamine at 37 °C in an atmosphere containing 5% CO_2_. The method used to quantify cell viability was the 3-(4.5-dimethylthiazol-2-yl)-2.5-diphenyltetrazolium bromide (MTT) assay as described by Gaspar et al. [40].

Briefly, the human tumor cells were cultivated in 96-well microplates in the supplemented medium and grown in an incubator with 5% CO_2_ at 37 °C until reaching 100% confluence, after the medium in each well was replaced by different concentrations of compounds’ solutions. The compounds were diluted to working solutions with the supplemented culture medium containing 0.5% DMSO final concentration to dissolve the compounds. Cells treated only with a medium containing 0.5% DMSO served as a negative control. All experiments were replicated eight times.

After 24 h incubation, the solutions were discarded and 100 µL of 0.5 mg/mL MTT in DMEM culture medium was added to each well. After incubating at 37 °C, 5% CO_2_ for 2 to 4 h, the solution was discarded and 100 µL of methanol was added to completely dissolve the formazan crystals. The absorbance at 595 nm (detection wavelength) against 630 nm (reference wavelength) was registered. For each concentration of the compounds, the percentage of growth inhibition/cytotoxicity was evaluated considering 100% of viability for the absorbance of the negative control. Dose–response curves for each compound were plotted using the Graph Pad Prim 4.0 software and several parameters were determined, namely the half-maximal growth inhibition concentration (EC_50_) and the Hill slope (which describes the steepness of the curve adjusted according to the Hill equation). Additionally, we determined the upper limit, that is, the concentration of the compounds where maximum growth inhibition/cytotoxicity occurs and the lower limit, that is the highest compound concentration showing the lowest growth inhibition below which the curves show a bottom plateau.

### 3.2. ADME Predictions

The ADME properties of compounds were predicted using SwissADME [34] and the pkCSM server [35]. ChemDraw 20.0 software was used to generate SMILES notations for all the synthesized molecules and reported chemical structures.

## 4. Conclusions

In conclusion, a library of 17 new thiobenzanilide derivatives was synthesized and all compounds were evaluated as potential anticancer agents against breast cancer and difficult-to-treat melanomas.

Results suggest that compound **17**, with an EC_50_ of 11.8 ± 0.2, and compounds **8** and **9**, are promising compounds to be further investigated to address the scourge of melanomas. As for the susceptibility of the MCF-7 cells to the thiobenzanilides studied in the present work, compound **15**, showed the highest cytotoxic potential, with an EC_50_ value of 43 µM, close to the EC_50_ value of 30 µM obtained for tamoxifen, the most common drug used in breast cancer therapy.

The in silico ADME properties of all compounds were also evaluated confirming the interesting profile of compound **8** with high predicted bioavailability and low hepatoxicity, whereas compounds **9**, **15** and **17** may be considered noteworthy scaffolds for further improvement.

## Figures and Tables

**Figure 1 molecules-28-01877-f001:**
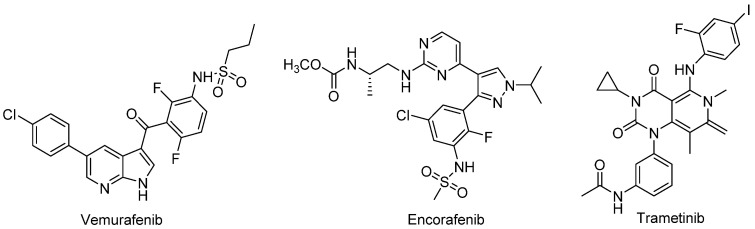
Structures of FDA-approved drugs for metastatic melanoma.

**Figure 2 molecules-28-01877-f002:**
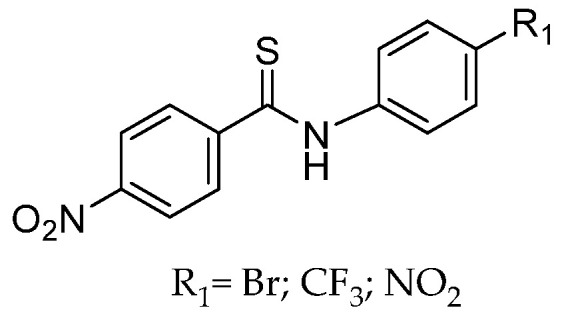
Thiobenzanilide derivatives active against melanoma A375 cell line [21].

**Figure 3 molecules-28-01877-f003:**
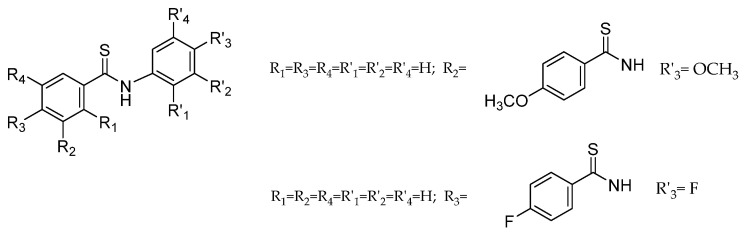
Structure of the thiobenzanilides showing antiproliferative activity against estrogen-dependent MCF-7 cancer cells [22].

**Figure 4 molecules-28-01877-f004:**
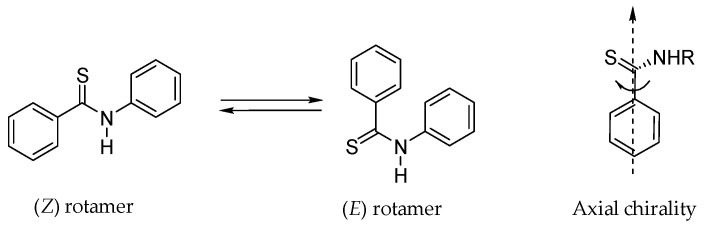
(*Z*) and (*E*) rotamers and axial chirality for thiobenzanilides.

**Figure 5 molecules-28-01877-f005:**
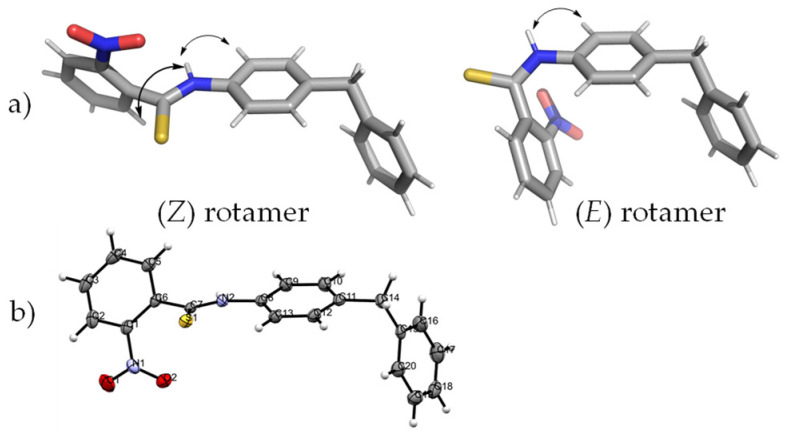
(**a**) (*Z*) and (*E*) rotamers of compound **1** and their NOESY correlation structures were obtained with PyMOL (**b**) ORTEP representation of compound **1**.

**Figure 6 molecules-28-01877-f006:**
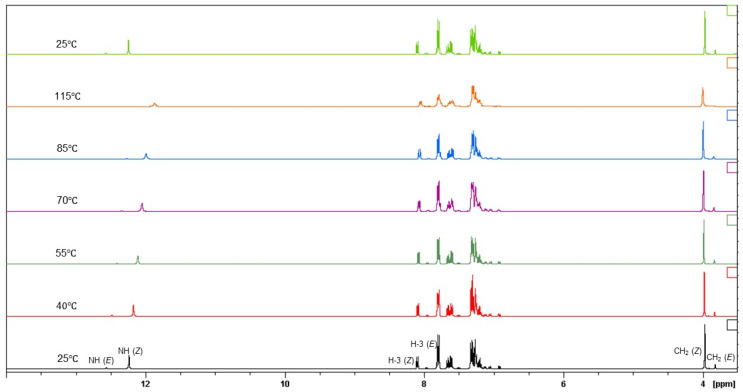
Variable temperature ^1^H NMR spectra (DMSO-*d*_6_, 25–115 °C) of compound **1** (zoom 3.5–14 ppm).

**Figure 7 molecules-28-01877-f007:**
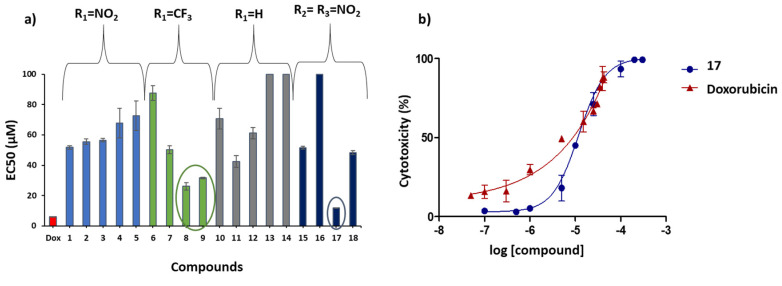
(**a**) Comparative compilation of all EC_50_ for A375 cell line of the tested thiobenzanilides. The dark blue circle indicates the most promising thiobenzanilide derivative and the green circle refers to the two second-best compounds; (**b**) Dose–response curves for compound **17** and doxorubicin.

**Table 1 molecules-28-01877-t001:** Synthetic pathway and structure of thiobenzanilides **1**–**18**.

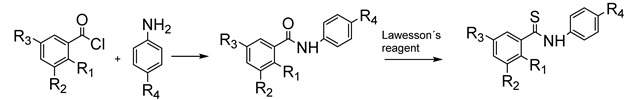
Compound	R_1_	R_2_	R_3_	R_4_	Yield (%)
**1**	NO_2_	H	H	CH_2_Ph	58
**2**	NO_2_	H	H	OPh	45
**3**	NO_2_	H	H	(CH_2_)_7_CH_3_	13
**4**	NO_2_	H	H	O(CH_2_)_7_CH_3_	15
**5**	NO_2_	H	H	H	15
**6**	CF_3_	H	H	(CH_2_)_7_CH_3_	37
**7**	CF_3_	H	H	O(CH_2_)_7_CH_3_	86
**8**	CF_3_	H	H	H	21
**9**	CF_3_	H	H	OPh	82
**10**	H	H	H	CH_2_Ph	69
**11**	H	H	H	OPh	56
**12**	H	H	H	(CH_2_)_7_CH_3_	42
**13**	H	H	H	O(CH_2_)_7_CH_3_	14
**14**	H	H	H	H	72
**15**	H	NO_2_	NO_2_	CH_2_Ph	91
**16**	H	NO_2_	NO_2_	(CH_2_)_7_CH_3_	74
**17**	H	NO_2_	NO_2_	O(CH_2_)_7_CH_3_	56
**18**	H	NO_2_	NO_2_	H	34

**Table 2 molecules-28-01877-t002:** The Gibbs energy difference between the *Z*/*E* conformers (ΔΔ*G*) and their (*Z*)/(*E*) population ratios obtained from a theoretical two-state model or from the ^1^H NMR experiments (only for Compound **1**) in different solvents at 25 °C.

Compound	Solvent	ΔΔ*G* (kcal/mol)	(*Z*)/(*E*)	Exp. (*Z*)/(*E*)
**1**	CDCl_3_	−0.55	1:0.40	1:0.6
**1**	C_6_D_6_	−0.75	1:0.28	1:0.4
**1**	THF-*d*_8_	−1.03	1:0.18	1:0.1
**1**	(CD_3_)_2_CO	−1.09	1:0.16	1:0.1
**1**	DMSO-*d*_6_	−1.01	1:0.18	1:0.1
**2**	CDCl_3_	−0.42	1:0.49	-
**3**	CDCl_3_	−0.60	1:0.36	-
**4**	CDCl_3_	−0.13	1:0.80	-
**5**	CDCl_3_	−0.60	1:0.36	-
**6**	CDCl_3_	−4.51	1:0.00	-
**7**	CDCl_3_	−1.62	1:0.07	-
**8**	CDCl_3_	−2.10	1:0.03	-
**9**	CDCl_3_	−1.64	1:0.06	-
**10**	CDCl_3_	−2.03	1:0.03	-
**11**	CDCl_3_	−1.35	1:0.10	-
**12**	CDCl_3_	−1.58	1:0.07	-
**13**	CDCl_3_	−2.16	1:0.03	-
**14**	CDCl_3_	−1.24	1:0.12	-
**15**	CDCl_3_	−0.65	1:0.34	-
**16**	CDCl_3_	−0.61	1:0.36	-
**17**	CDCl_3_	−0.54	1:0.40	-
**18**	CDCl_3_	−1.20	1:0.13	-

**Table 3 molecules-28-01877-t003:** EC_50_ values (μM) and respective standard deviations (*s*) obtained for compounds **1**−**18** for MCF-7 and A375 cell lines, after incubation at 37 °C for 24 h.

	A375				MCF-7
Compound	${\mathrm{EC}}_{50}\;\pm$ s (μM)	Upper Limit (μM)	Lower Limit (μM)	Hill Slope	${\mathrm{EC}}_{50}\;\pm$ s (μM)
**1**	52 ± 1	200.0	2.0	9.6	>100
**2**	56 ± 2	200.0	1.0	3.7	>100
**3**	57 ± 1	200.0	1.0	5.4	68 ± 1
**4**	68 ± 9	300.0	1.0	4.6	65 ± 2
**5**	73 ± 9	200.0	10.0	2.7	>100
**6**	88 ± 5	300.0	10.0	3.8	90 ± 2
**7**	50 ± 3	300.0	0.5	2.4	>100
**8**	26± 3	200.0	0.5	1.4	>100
**9**	31.6 ± 0.6	200.0	1.0	2.9	73 ± 2
**10**	71 ± 7	300.0	0.5	1.9	>100
**11**	42± 4	300.0	1.0	2.0	>100
**12**	61 ± 4	250.0	0.5	2.4	>100
**13**	>100	-	-	-	>100
**14**	>100	-	-	-	>100
**15**	51.6 ± 0.9	200.0	1.0	6.9	43 ± 2
**16**	>100	-	-	-	>100
**17**	11.8 ± 0.2	300.0	0.1	1.5	96 ± 2
**18**	48 ± 1	200.0	2.0	6.0	93 ± 3
Doxorubicin	6.0 ± 0.2	-	-	-	-
Tamoxifen	-	-	-	-	30.0 ± 0.6

## Data Availability

Not applicable.

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
