# Peer review of "Synthesis and the In Vitro Evaluation of Antitumor Activity of Novel Thiobenzanilides"

_molecules, 2023, doi:10.3390/molecules28041877_

Round 1

Reviewer 1 Report

Álvaro-Martins et al. present anticancer properties of novel thiobenzanilides in cancer types such as melanomas and breast cancer cells.

The anticancer effects of thiobenzanilides are already reported in the literature, however, the authors claims derivatives of thiobenzanilides with better and specific efficacies.

This paper has limited merit to understand and explore the anticancer effects of novel thiobenzanilides. At the same time, this paper requires additional data and discussion.

Major points that may enhance the quality of this manuscript.

1. First, authors should make clear rationale on the selection of melanoma and breast cancer cells. Whether selection of these cancer cell lines are based on the genetic or drug responsiveness profiles.

2. These novel thiobenzanilides should explored to assess whether antiproliferative or apoptotic cell death. Only MTT assay may not be enough to substantiate claims.

3. These cancer cells treated by novel thiobenzanilides, authors may need to show the presence of these drugs at the intracellular levels.

4. The conclusion stated as “Overall, this work discloses new compounds that are worth to be considered as new leads for fur-31 ther development of future therapeutic agents, being specially promising for skin cancer therapy” is not in alignment with fact that two different cell lines are used including for skin cancer and breast cancer.

5. The authors may focus on the skin cancer by opting two cell lines and study in details whether antiproliferative or apoptotic cell death.

6.ADMET profile of novel thiobenzanilides should be compared with already reported thiobenzanilides anticancer drugs.

7. Please include relevant reference to substantiate claims and discussion.

8. Flow of discussion needs improvement.

Reviewer 2 Report

The authors of this article report a set of new thiobenzanilides, their synthesis and the in vitro analysis of antitumor properties. Indeed, thiobenzanilides disclose strong biological activity as antimycotic and antifungal agents. Moreover, recently, it was demonstrated that these complexes exhibit robust cytotoxicity. The authors carefully describe the synthesis and description of these compounds, evaluate their cytotoxicity in vitro in human cancer cell lines A375 melanoma and MCF-7 breast cancer. The usage of English is impeccable. This manuscript can be published after the authors address several important issues:

1. The claims about the correlation between the presence of electron withdrawing groups in the compounds and their cytotoxicity is a gross speculation. The authors need to complete an in-depth mechanistic study of interaction of the newly synthesized compounds with their biomolecular targets, only afterwards, one could elaborate on such an important issue. That is why these claims should be eliminated.

2. The energy minimization with MM2 software is not enough. The force field cannot adequately incorporate the electronic and polarization effects, it can be used only as a tool for obtaining a set of possible molecular conformations which should be subsequently minimized by quantum chemical methods. It is essential to calculate the Gibbs free energies and dipolar moments of compounds via a quantum chemical approach (DFT). It is suggested for the authors:

(a) to perform conformer searches for the complexes using either MM2 or CREST (https://crest-lab.github.io/crest-docs/) or any other tool,

(b) optimize the lowest 5-10 conformers by DFT in several solvents, thus, the authors will be able to describe the variation in the rotamer’s population, and its dependence on the dielectric properties of solvents.

3. The authors discuss the presence of electron-withdrawing/donating groups in the compounds. Again, there is no quantitative analysis here. Meanwhile, the simple DFT optimizations will yield Mulliken populations, so the authors will be able to quantify the presence of charges on various atoms.

The thiobenzanilide compounds studied by the authors are very small and easy to calculate at QM level. I think such a calculation will greatly increase the value of this manuscript.

Reviewer 3 Report

Authors:

The series of compounds you synthesized is quite interesting, however, their cytotoxicity is rather low. Nevertheless, your work should be published after correcting several error I have found.

The introduction contains all necessary items of information. The experimental part was well planned and conducted, and the Results and Conclusion are well understandable. Below, you will find several items to be corrected.

Abstract, lines 25, 29 and 30: EC50 should be written as EC50.

Line 102: Correct the error.

Table 1: The reaction scheme therein appears as not completed. Please, correct.

Figure 6: 1H RMN should be written as 1H NMR.

Lines 175 and 177: Please, correct grammatics.

I recommend minor revision before this manuscript is accepted for publication.

Reviewer 4 Report

This manuscript reported the synthesis of 18 thiobenzanilide derivatives, and the antitumor activity of thiobenzanilide derivatives was evaluated in vitro. The thiobenzanilide derivatives were synthesized by varying the substitution pattern of the acyl and aniline moieties. The population of the rotamers in different solvents was studied, and the effect of solvent properties on the rotamer’s population was discussed. The co-existence behavior of thiobenzanilide derivatives at different temperatures was also studied. Furthermore, the effect of different concentrations of thiobenzanilides to inhibit A375 and MCF-7 cell growth were investigated, and the effects of substituents on cytotoxicity were analyzed. I think this manuscript is suitable for publication in Molecules, provided the authors consider the following comments.

(1)    The authors mentioned that but recent advances in the so-called new therapies, including immunotherapy and targeted drugs, increased the five years’ survival rate of these patients by 52 % [4], which is, however, still a very disappointing number (Page 2). The advantages and disadvantages of new therapies need to be further illuminated, highlighting the importance of anticancer drugs.

(2)    In the fourth paragraph (Page 2), the authors described the challenges of developing cancer therapeutics, but the importance of developing novel drug molecules were lacked. The corresponding supplement and modification are suggested.

(3)    To make the manuscript more convincing, the photographs of the cancer cells before and after treatment with the thiobenzoimide derivative should be provided.

(4)    The antiproliferation ability of thiobenzoimide derivatives, doxorubicin and tamoxifen was studied (Table 3, Page 7), but the advantages of the derivatives were not illuminated. The anti-tumor advantages of the derivatives should be highlighted compared with doxorubicin and tamoxifen.

(5)    More relevant works about tumor treatment should be cited such as Ji et al., Angew. Chem. Int. Ed., 2022, e202200237; Chen et al., Adv. Funct. Mater., 2020, 30, 1909806.

Round 2

Reviewer 1 Report

The authors have attempted to address comments substantially. However, this work needs better future experimentations. 

Reviewer 2 Report

The authors have accurately addressed all the issues brought up by me during the last round of peer-review, thus substantially improving the manuscript. The claims about the correlation between the availability of electron withdrawing groups and the cytotoxicity of complexes were eliminated, whereas the DFT optimizations were performed, yielding an agreement between the computed and experimental data. The manuscript can be published in its present form.